# Proposed Standard Test Protocols and Outcome Measures for Quantitative Comparison of Emissions from Electronic Nicotine Delivery Systems

**DOI:** 10.3390/ijerph19042144

**Published:** 2022-02-14

**Authors:** Edward C. Hensel, Nathan C. Eddingsaas, Qutaiba M. Saleh, Shehan Jayasekera, Samantha Emma Sarles, A. Gary DiFrancesco, Risa J. Robinson

**Affiliations:** 1Department of Mechanical Engineering, Rochester Institute of Technology, Rochester, NY 14623, USA; agdpci@rit.edu (A.G.D.); rjreme@rit.edu (R.J.R.); 2School of Chemistry and Materials Science, Rochester Institute of Technology, Rochester, NY 14623, USA; ncesch@rit.edu; 3Electrical and Computer Engineering, Rochester Institute of Technology, Rochester, NY 14623, USA; qms7252@rit.edu; 4Mechanical and Industrial Engineering, Rochester Institute of Technology, Rochester, NY 14623, USA; gbj6142@rit.edu; 5Biomedical Engineering, Rochester Institute of Technology, Rochester, NY 14623, USA; ses9066@rit.edu

**Keywords:** e-cigarette, regulatory science, substantial equivalence, public health, nicotine, tobacco product comparison

## Abstract

This study introduces and demonstrates a comprehensive, accurate, unbiased approach to robust quantitative comparison of electronic nicotine delivery systems (ENDS) appropriate for establishing substantial equivalence (or lack thereof) between inhaled nicotine products. The approach is demonstrated across a family of thirteen pen- and pod-style ENDS products. Methods employed consist of formulating a robust emissions surface regression model, quantifying the empirical accuracy of the model as applied to each product, evaluating relationships between product design characteristics and maximum emissions characteristics, and presenting results in formats useful to researchers, regulators, and consumers. Results provide a response surface to characterize emissions (total particulate matter and constituents thereof) from each ENDS appropriate for use in a computer model and for conducting quantitative exposure comparisons between products. Results demonstrate that emissions vary as a function of puff duration, flow rate, e-liquid composition, and device operating power. Further, results indicate that regulating design characteristics of ENDS devices and consumables may not achieve desired public health outcomes; it is more effective to regulate maximum permissible emissions directly. Three emissions outcome measures (yield per puff, mass concentration, and constituent mass ratio) are recommended for adoption as standard quantities for reporting by manufacturers and research laboratories. The approach provides a means of: (a) quantifying and comparing maximal emissions from ENDS products spanning their entire operating envelope, (b) comparative evaluation of ENDS devices and consumable design characteristics, and (c) establishing comparative equivalence of maximal emissions from ENDS. A consumer-oriented product emissions dashboard is proposed for comparative evaluation of ENDS exposure potential. Maximum achievable power dissipated in the coil of ENDS is identified as a potentially effective regulatory parameter.

## 1. Introduction

### 1.1. Context of Prior Work

Electronic nicotine delivery systems (ENDS), also called electronic cigarettes or vaping devices, have emerged globally as a leading technology for users to inhale nicotine, and have displaced significant market share from traditional combustible cigarettes. There is a significant public health need to have a robust, repeatable, and statistically valid means of assessing the potential adverse public health effects of ENDS use. Furthermore, there is a need to conduct comparisons of harm potential between ENDS products as a key element of the regulatory approval of proposed new products. An essential barrier to conducting such comparisons is the lack of standardized ENDS emissions reporting measures across research labs. Decades of experience with combustible cigarettes have demonstrated the yield of toxicants from cigarettes is jointly dependent upon both the constituents present in the cigarette and how it is smoked, commonly referred to as the “puffing topography”. Similarly, ENDS emissions are dependent upon the design and materials of the ENDS and the constituents present in the consumable e-liquid of the ENDS. This study proposes a baseline “emissions model” and outcome measures for comparing the relative harm potential between ENDS products. 

As detailed previously [1,2], no widely accepted puffing topographies or emissions outcome measures have been established for use in ENDS emission studies, although emissions have clearly been demonstrated to be dependent upon usage conditions and product characteristics. A proposed vaping machine standard [3] does not reflect the range of use behavior associated with product use but offers a consistent foundation for comparative emissions studies between ENDS. Relatively few emissions results and product characterizations have been reported in the literature in a format which enables a side-by-side comparison between ENDS, such as would be required to establish statistically significant substantial equivalence. While ENDS regulation varies around the world, current US regulations require manufacturers to submit a premarket tobacco product application (PMTA) containing an emissions characterization of the product, among other items. A critical aspect of product review and clearance for marketing is whether the product may be deemed substantially equivalent to a predicate product already in the marketplace, and whether the proposed product may have comparatively adverse public health consequences. This article focuses on providing quantitative measures for comparison of emissions between inhaled nicotine products.

The variety of emission outcome measures and underlying operating conditions of the ENDS reported in the literature make it difficult to compare results between laboratories and studies. The recent article by Son et al. [1] presented emissions data from four ENDS devices. Of the four devices reported, one overlaps with results to be presented here, the JUUL pod-style ENDS. Son reported the nicotine yield as 0.390 ± 0.0305 [mg/puff] combined across four e-liquids (JUUL brand names: Virginia Tobacco, Fruit Melody, Crème Brulee, and Cool Mint) with a manufacturer-reported nicotine concentration of nominally 59 [mg/mL]. Son reported that nicotine was collected from the aerosol generated by five puffs of the device and captured on a glass fiber filter pad using puff volumes from 67 to 133 [mL] and puff durations from 4 to 5 [s]. The mass of condensed aerosol emissions collected from ENDS emissions is commonly referred to as yield of total particulate matter (TPM). The data provided Son et al. [1] contain an excellent summary of emission yield per puff (TPM, carbon monoxide, formaldehyde, acetaldehyde, acetone, acrolein, propionaldehyde, crotonaldehyde, 2-butanone, methacrolein, n-butyraldehyde, benzaldehyde, valeraldehyde, glyoxal, m-tolualdehyde, and hexaldehydes) from four ENDS spanning five to seven flow conditions per device. For comparison purposes, the total mass yield per puff from JUUL reported by Son et al. is provided in Table 1. The TPM yield per puff values in Table 1 provide data points for comparison with results presented herein. Son’s study reported most of the outcome measures in terms of yield per puff, with the exception of carbon monoxide, which was reported as the ratio of CO to nicotine. Unfortunately, nicotine yield per puff was not reported as a function of flow conditions. Yield measures (mass per puff) of TPM and nicotine reported in Son et al. are directly comparable to values reported here. While the current work does not report yield of additional harmful and potentially harmful constituents (HPHCs) per puff, Y_HPHC_, this outcome measure reported by Son et al. is a valuable measure for reporting one aspect of ENDS emissions and can be used to compare the change in relative exposure a user may experience by switching from one ENDS product to another. 

### 1.2. Study Objectives

The first objective of this study is to demonstrate accurate, unbiased emission models (EM) for quantifying emissions generated from a variety of pen- and pod-style ENDS across their respective operating envelope in terms of four standard emissions outcome measures.

The second objective is to identify maximum response characteristics of the EM which may be used to compare relative emissions between products (presented as an emissions characteristics dashboard). 

The third objective is to investigate associations between the maximal EM characteristics and underlying product design characteristics.

The fourth objective is to propose those product characteristics likely to be effective regulatory parameters to achieve positive outcomes related to public health, reduction in harm potential, and clinical guidance.

## 2. Materials and Methods 

### 2.1. Experimental Materials, Methods, and Data Set

Prior work [2] described the materials and methods used for gathering experimental data previously reported. Briefly, test specimens were procured for testing 13 pen- and pod-style ENDS products from commercial vendors. Products were selected based on their popularity in the US market, being legally available for sale in the State of New York, having no user-adjustable power settings, and including pen- and pod-style ENDS, disposable and refillable products, and button- and flow-activated devices. The prior publication reported full details about each product tested, including observed manufacturing variation in product characteristics, the experimental conditions of emissions testing, and the yield associated with each experimental trial. An emissions screening protocol consisted of two series of machine puffing trials, referred to as the “activation duration” family and “activation flow rate” family of trials. Each trial consisted of nominally 50 homogenous puffs, at the conclusion of which the mass decrease of the ENDS and the mass increase of TPM collected on a Cambridge filter pad were measured gravimetrically. Each trial was conducted with the ENDS oriented at an inclination angle of 30 degrees to ensure that e-liquid could not be gravity fed from the reservoir into the emissions collection system. The coil resistance of each product was measured using an unbiased four-lead method [4,5] before and after the emissions trials. The propylene glycol to glycerin ratio (PG:GL) of the unpuffed e-liquid was measured using NMR and the nicotine mass ratio of both the unpuffed e-liquid, *f_Nic, Unpuffed_*, and aerosol collected on each pad, f_Nic_, was determined using GC-MS. The nicotine mass ratio is defined as the ratio of mass nicotine collected on the pad to the total mass collected and was reported to be largely independent of both flow rate and duration for the 13 low-power ENDS studied. The gathered data were analyzed, and the operating envelope of each ENDS device was characterized in terms of four parameters: the minimum aerosolization duration, MinAD, was the lowest puff duration below which no measurable TPM was reliably detected. The maximum aerosolization duration, MaxAD, was the duration above which the TPM per puff stopped increasing, typically indicative of a “power off” time limit in the ENDS control logic. The minimum aerosolization flow rate, MinAF, was the flow rate below which the ENDS did not reliably activate and did not produce detectable TPM. The maximum aerosolization flow rate, MaxAF, represents the highest flow rate at which ENDS aerosol was produced without aspirating e-liquid directly into the flow path. The maximum relative nominal power dissipated in the ENDS coil is denoted as Max P_Nom_, and reflects the highest nominal operating power of the device. The 13 devices studied herein did not offer user-adjustable power features. In the event of adjustable power ENDS, the value reported should reflect the maximum achievable value. All underlying data and computed results were published previously [2]. The current work focuses on developing a convenient mathematical representation of emissions across a variety of pen- and pod-style ENDS appropriate for use in conjunction with the previously described [6] and validated [7] behavior-based yield model. Each product configuration tested was referred to with a unique identifier in the form of EC*xx-yy*, where *xx* referred to a particular PCU configuration (e.g., a specific coil chosen for the emissions test series, along with the manufacturer name and model number) and *yy* referred to consumable (i.e., e-liquid PG:GL ratio, nicotine concentration, flavor, manufacturer, and brand name). The product design characteristics of all products tested are summarized in Table 2.

### 2.2. Emissions Outcome Measures 

We propose four standard emissions outcome measures. The first is the yield of total particulate matter (condensed aerosol) per puff, *Y_TPM_*, [mg TPM/puff]. The second outcome measure is TPM mass concentration, C_TPM_, [mg TPM/mL]. Each constituent present in the aerosol is characterized with two outcome measures. For nicotine, these are the nicotine mass ratio and yield, *f_Nic_* [mg Nic/mg TPM] and *Y_Nic_* [mg Nic/puff], respectively. Any other HPHC present in the aerosol may similarly be characterized by its mass ratio *f_HPHC_* [mg HPHC/mg TPM] and yield *Y_HPHC_* [mg HPHC/puff]. Son et al. [1] reported the yield of TPM and several HPHCs as a function of puff flow rate and duration for multiple ENDS, which permits calculation of mass ratio for each. This demonstrates how the proposed outcome measures facilitate reuse of results and data between independent laboratories. *Y_Nic_* and corresponding values of *Y_HPHC_* provide a direct comparison of toxicant exposure per puff between ENDS products. However, the yield per puff values alone does not identify whether the exposure to a particular *HPHC* is a result of a change in the *TPM* yield per puff, or whether the *HPHC* yield per puff results from some other characteristic of the nicotine delivery product. The values of *f_Nic_* and *f_HPHC_* normalize the toxicant yield to the *TPM* yield and provide insight into changes in HPHC emissions which may be evident over the operating envelope of the device. In combination with the reporting of *Y_TPM_* and *C_TPM_*, the values of f_HPHC_ permit ready calculation of both *Y_HPHC_* and *C_HPHC_*. In short, any two of the outcome measures (Y, C, and f) permit calculation of the third. However, reporting only one of the three outcome measures introduces ambiguity which may be exploited by parties seeking to circumvent product emissions regulations. Each outcome measure may be interpreted as a proxy for measure of relative exposure, and subsequently harm potential, between nicotine delivery products. *Y_TPM_* is a measure of the total aerosol exposure delivered to a user’s mouth, without articulating the constituents present in the aerosol. The mass ratio, *f_HPHC_*, of any constituent is a measure of the relative contribution of any particular *HPHC* to the total aerosol yield. Thus, by studying how *f_HPHC_* changes as a function of product characteristics, we can investigate which aspects of a product give rise to concerning levels of a particular *HPHC*, and whether those concerns exist over the entire envelope. Reporting *HPHC*s only as yield obfuscates whether a change in *HPHC* emissions is due to the variation in *TPM* yield or is due to some change in the ENDS emissions characteristics.

### 2.3. Empirical TPM Yield Emission Model

An empirical emission model of the *TPM* yield per puff is defined herein as the product of logistic (sigmoid) functions of puff flow rate and puff duration as shown in Equation (1), identically zero below MinAF and MinAD as given by Equation (2), and limited to the boundary condition above MaxAF and MaxAD as defined by Equations (3)–(5), consistent with a constant surface extrapolation:(1)$$\begin{array}{c} \begin{array}{c} {\hat{Y}}_{TPM}=\frac{1}{min\left(B_{1},B_{4}\right)}\left(\frac{B_{1}}{1+e^{-B_{2}\left(q-B_{3}\right)}}\right)\left(\frac{B_{4}}{1+e^{-B_{5}\left(d-B_{6}\right)}}\right)\;\forall \;MinAF\leq q\leq MaxAF;MinAD\leq d\leq MaxAD\; \end{array} \end{array}$$
(2)$$\begin{array}{c} \begin{array}{c} {\hat{Y}}_{TPM}=0\;\;\;\;\;\;\;\;\;\;\;\;\;\;\;\;\;\;\;\;\;\;\;\;\;\;\;\;\;\;\;\;\;\;\;\;\;\;\;\;\;\;\;\;\;\;\;\;\;\;\;\forall \;\;q<MinAF;\;\;d<MinAD\; \end{array} \end{array}$$
(3)$$\begin{array}{c} \begin{array}{c} {\hat{Y}}_{TPM}=\frac{1}{min\left(B_{1},B_{4}\right)}\;\left(\frac{B_{1}}{1+e^{-B_{2}\left(q-B_{3}\right)}}\right)\left(\frac{B_{4}}{1+e^{-B_{5}\left(MaxAD-B_{6}\right)}}\right)\;\;\forall \;\;MinAF\leq q\leq MaxAF;d>MaxAD\; \end{array} \end{array}$$
(4)$$\begin{array}{c} \begin{array}{c} {\hat{Y}}_{TPM}=\frac{1}{min\left(B_{1},B_{4}\right)}\;\left(\frac{B_{1}}{1+e^{-B_{2}\left(MaxAF-B_{3}\right)}}\right)\left(\frac{B_{4}}{1+e^{-B_{5}\left(d-B_{6}\right)}}\right)\;\;\forall \;\;q>MaxAF;\;MinAD\leq d\leq MaxAD\; \end{array} \end{array}$$
(5)$$\begin{array}{c} \begin{array}{c} {\hat{Y}}_{TPM}=\frac{1}{min\left(B_{1},B_{4}\right)}\;\left(\frac{B_{1}}{1+e^{-B_{2}\left(MaxAF-B_{3}\right)}}\right)\left(\frac{B_{4}}{1+e^{-B_{5}\left(MaxAD-B_{6}\right)}}\right)\;\;\forall \;\;q>MaxAF;\;d>MaxAD\; \end{array} \end{array}$$
where B_3_ and B_6_ are the centroids of the logistic function of puff flow rate and duration, respectively, B_2_ and B_5_ are the logistic growth rates for puff flow rate and duration, respectively, B_1_ and B_4_ are the maximum model response for puff flow rate and duration, respectively, and ${\hat{Y}}_{TPM}$ is the model-predicted value of the yield per puff given a puff flow rate, q, and duration, d. The leading quotient permits the maximum TPM yield to be limited either by flow rate or by puff duration, appropriate to the observed response of each ENDS. The model is C_0_ continuous, guarantees the ${\hat{Y}}_{TPM}$ surface remains flat at long durations, d > MaxAD, when the coil becomes de-energized, and the mass concentration C_TPM_ decays with increasing puff volume above MaxAD and MaxAF. Using the experimental methods described previously [2], an L-vector of Y_TPM_ observations was collected as a function of MinAF < q < MaxAF while d was held constant and another M-vector of Y_TPM_ observations was collected as a function of MinAD < d < MaxAD while q was held constant, for a total of L + M data points. The unweighted Levenberg–Marquardt nonlinear least squares algorithm [8,9] was used to estimate the parameters B_1_ through B_6_ for each ENDS investigated. Coefficients B_1_ through B_3_ were determined by nonlinear regression of the “activation flow rate” data while B_4_ through B_6_ were determined by nonlinear regression of the “activation duration” data. The product of the two regression formulas, Equation (6), was then used to compute the sum of the squared residuals between the experimental observations and the hybrid model predictions:(6)$$SSR=\sum_{i=1}^{L+M}{\left(Y_{TPM}-{\hat{Y}}_{TPM}\right)}^{2}$$

The initial guesses, B_0_, for the parameters to initialize the nonlinear least squares algorithm were taken to be: B_0,1_ = max(*Y_TPM_*) from the “activation flow rate” series, B_0,2_ = 16 π/(MaxAF − MinAF), B_0,3_ = MinAF + 0.1 (MaxAF − MinAF), B_0,4_ = max(*Y_TPM_*) from the “activation duration rate” series, B_0,5_ = π/(MaxAD − MinAD), and B_0,6_ = MinAD + 0.5 (MaxAD − MinAD).

### 2.4. Empirical TPM Concentration Emission Model

In addition to quantifying mass yield per puff of TPM, the model may be used to quantify the mass concentration of aerosol, denoted ${\hat{C}}_{TPM}$. The mass concentration is yield per puff normalized by the volume of the puff, Equation (7):(7)$$\begin{array}{c} \begin{array}{c} \;\;{\hat{C}}_{TPM}\left(q,d\right)=\frac{{\hat{Y}}_{TPM}\left(q,d\right)}{q\cdot d}\; \end{array} \end{array}$$

### 2.5. Empirical Nicotine Mass Fraction Model

The constituent mass ratio of nicotine, or any other aerosol constituent of interest, may be dependent upon operating parameters such as coil power, temperature, and e-liquid composition in addition to user topography as shown in Equation (8). Investigating the mass ratio of constituents as a function of flow conditions enables research laboratories to leverage the work of one another. For the 13 ENDS studied herein, *f_Nic_* was determined to be independent of flow rate and duration, and the ENDS devices had no user-adjustable parameters, such as power. Therefore, the nicotine mass ratio surface was simplified to the mean of the empirical observations of the nicotine mass ratio observed in the aerosol samples:(8)$$\begin{array}{c} \begin{array}{c} {\hat{f}}_{Nic}=\;B_{7}+B_{8}q+B_{8}q+B_{10}\;qd+\cdots \approx {\bar{f}}_{Nic}\; \end{array} \end{array}$$

### 2.6. Empirical Nicotine Yield Model

The nicotine yield per puff was estimated as a separable model of the nicotine mass ratio and the TPM yield per puff using an implicit assumption of linear superposition. For the current family of products tested, the nicotine mass ratio was observed independent of flow rate and duration as shown in Equation (9):(9)$$\begin{array}{c} \begin{array}{c} \;\;{\hat{Y}}_{Nic}\left(q,d\right)={\hat{f}}_{Nic}\left(q,d\right)\;\;{\hat{Y}}_{TPM}\left(q_{n},d_{n}\right)\approx \;{\hat{f}}_{Nic}\;{\hat{Y}}_{TPM}\left(q_{n},d_{n}\right) \end{array} \end{array}\;$$

### 2.7. Confidence Intervals on the Model Estimates

The standard error of the regression, *S_YTPM_*, is a measure of the agreement between the model-predicted values, ${\hat{Y}}_{TPM}$, and the *L* + *M* experimental observations, $Y_{TPM}$, given by [10].
(10)$$\begin{array}{c} \begin{array}{c} S_{YTPM}=\frac{1}{\left(L+M\right)}\sum_{i=1}^{L+M}{\left(Y_{TPM}-{\hat{Y}}_{TPM}\right)}^{2} \end{array} \end{array}$$

We estimated the 95% confidence interval on the estimate of *TPM* yield per puff as being *ε_YTPM_* ≈ ±1.96 *S_YTPM_*, and similarly estimated the uncertainty in the estimate of the *TPM* mass concentration, *C_TPM_*, as *ε_CTPM_* ≈ ±1.96 *S_CTPM_*; nicotine mass ratio, f_Nic_, as *ε_FNic_* ≈ ±1.96 *S_FNic_*; and nicotine yield as *ε_YNic_* ≈ ±1.96 *S_YNic_*. Each quantity for the standard error of the regression was computed for the actual number of empirical observations across flow rate, q, and duration, d, for the respective outcome measure. For example, the number of observations of f_Nic_ may differ from those of *Y_Nic_*. This approach was used as a means of assessing the impact of algebraic manipulation of the model, such as Equation (7), and the validity of the linear superposition assumption implied by Equation (9).

### 2.8. Presentation of Results

Several figures were generated for each ENDS screening emission model to illustrate the characteristics of the device. The first figure consisted of a scatter and regression plot of the “activation flow rate” data to assess the quality of parameters B_1_ through B_3_. The second figure was a scatter and regression plot of the “activation duration” data to assess the quality of parameters B_4_ through B_6_. The third figure consisted of a surface plot of the *TPM* yield per puff ${\hat{Y}}_{TPM}$ model predictions given by Equation (1) overlaid with the experimental data, *Y_TPM_*, as a function of puff flow rate, q, and duration, d. The fourth figure was a surface plot of the aerosol nicotine mass fraction, ${\hat{f}}_{Nic}$*,* given by Equation (8), which was taken to be a flat surface for each of the 13 products investigated. The fifth figure generated for each ENDS device consisted of a surface plot of the TPM mass concentration, ${\hat{C}}_{TPM}={\hat{Y}}_{TPM}$/(q∙d), model predictions overlaid with the experimental data, *C_TPM_*. The sixth figure consisted of the nicotine yield per puff using Equation (9). Each model is described in terms of the six parameters B_1_–B_6_, *p* values on each, coefficient of determination, R^2^, and the 95% confidence interval on the regression for each of the activation flow rate and activation duration series. The accuracy of the surface models of *Y_TPM_*, *C_TPM_*, and *Y_Nic_* was quantified as the 95% confidence interval for the model computed from the corresponding standard error of regression analogous to Equation (10). All figures and details associated with each product tested are presented as Appendix A accompanying this article.

The screening emissions model of each device was summarized in a table, appropriate for use in other computer programs. The peak values of each response surface were determined and denoted as Max ${\hat{Y}}_{TPM}$, Max ${\hat{C}}_{TPM}$, Max ${\hat{f}}_{Nic},$ and Max ${\hat{Y}}_{Nic}.$ This summary information for each product was then compiled into a “screening emissions model dashboard” to foster quantitative relative comparison of emissions between products.

### 2.9. Investigation of Association between Product Characteristics and Emissions Characteristics

Each of the four emissions characteristics (Max ${\hat{Y}}_{TPM}$, Max ${\hat{C}}_{TPM}$, Max ${\hat{f}}_{Nic},$ and Max ${\hat{Y}}_{Nic}$) was separately evaluated using multivariate linear regression to assess potential correlation between emissions and the product design and operating envelope characteristics listed in Table 2 as indicated by Equation (11):(11)$$\begin{array}{c} \begin{array}{c} Max\;{\hat{Y}}_{TPM}=\beta_{0}+\beta_{1}\;f_{Nic,ELiq}+\beta_{2}f_{PG,ELiq}+\beta_{3}R_{Coil}+\beta_{4}\;MaxP_{Nom}+\beta_{5}MaxAF+\beta_{6}\;MaxAD. \end{array} \end{array}$$

The six linear regression coefficients were computed using QR decomposition [11] including M-estimation to formulate estimating equations [12] which were solved using the method of iteratively reweighted least squares [13,14] as implemented in the commercially available fitlm package [9]. The value of each regression coefficient, *β*, and its corresponding sum of squared errors, SSE, test statistic, and p value were reported. The model (11) implicitly assumes the solvent is a binary mixture of propylene glycol and glycerin characterized by a single parameter, *f_PG_*, with nicotine added to the unpuffed e-liquid reported as a mass ratio *f*_*Nic*,*ELiq*_. More complex solvents would require additional characterization parameters. The mean value of coil resistance, *R_Coil_*, was taken from the previous report, as were the operating envelope parameters *MaxAF* and *MaxAD*.

The actual transient power was not measured for these experimental conditions. A constant voltage of *V_Nom_* = 3.3 [VDC] was assumed as a basis for conducting relative nominal maximum power comparisons between products. The maximum relative nominal power dissipated was determined using Equation (12):(12)$$\begin{array}{c} \begin{array}{c} Max\;P_{Nom}\equiv \;\frac{V_{Nom}^{2}}{R_{Coil}}\; \end{array} \end{array}$$

Equation (12) neglects any active control (such as constant voltage, constant current, or pulse modulation) which may be employed in the power control unit of the product. While not an accurate representation of the true power dissipated in the coil, the maximum nominal power is a reasonable method for conducting a first-order quantitative comparison between products and is representative of the maximum nominal power anticipated for the product.

After each multivariate linear regression, similar to Equation (11), was established for each of the four emissions characteristics, an added variable assessment based on the Frisch–Waugh–Lovell theorem [15] of each product characteristic was conducted to determine whether each adjusted product characteristic exhibited an effect on the adjusted emission characteristic which was statistically distinguishable from the mean adjusted response within 95% confidence bounds. Those product characteristics observed to be significant were recommended for regulatory consideration.

## 3. Results

### 3.1. Exemplary Results for One Product

An exemplary sequence of six figures for a single product (EC14-01 BLU myBlu PCU with BLU Classic Tobacco E-Liquid) is presented to illustrate the methods employed. A summary table of model parameters and fit quality estimates will be presented. Figure 1 shows the experimental data (markers) for TPM yield per puff as a function of flow rate while the nominal puff duration was held constant. The logistic regression model described by parameters B_1_, B_2_, B_3_ is shown as the solid line, while the 95% confidence interval on the regression is shown as dashed lines. Product EC14-01 exhibits a distinct minimum flow rate below which the coil does not activate and above which the coil consistently energizes.

Figure 2 shows the experimental data (markers) for TPM yield per puff as a function of duration while the nominal puff flow rate was held constant. The logistic regression model described by parameters B_4_, B_5_, B_6_ is shown as the solid line, while the 95% confidence interval on the regression is shown as dashed lines. Product EC14-01 exhibits a nonlinear relationship between TPM yield and puff duration, and the coil was observed to remain energized at least until 10 [s] puff duration, the maximum duration investigated here.

Figure 3 shows the hybrid emissions screening model for TPM yield per puff, ${\hat{Y}}_{TPM},$ as a function of flow rate and duration with the surface as defined by Equations (1) through (5). The underlying data are illustrated by the markers, while the semitransparent surfaces reflect the 95% confidence interval, ε_YTPM,_ associated with the standard error of the regression, *S_YTPM_*, given by Equation (10). The value of Max ${\hat{Y}}_{TPM}$ was observed to be 23.3 ± 2.71 [mg/puff] for product EC14-01.

Figure 4 shows the hybrid emissions screening model for TPM mass concentration as a function of flow rate and duration with the surface as defined by Equation (7). The underlying data are illustrated by the markers, while the semitransparent surfaces reflect the 95% confidence interval, ε_CTPM_. The value of Max ${\hat{C}}_{TPM}$ was observed to be 0.110 ± 0.023 [mg/mL] for product EC14-01.

Figure 5 illustrates that no statistically significant dependence of f_Nic_ was observed as a function of puff flow rate or duration. While true for all 13 products tested here, the method permits topography dependence of all constituents. In general, a multivariate model such as that shown by Equation (8) may be more appropriate. The underlying data are illustrated by the markers, while the semitransparent surfaces reflect the 95% confidence interval, ε_fNic_. The value of Max ${\hat{f}}_{Nic}$ was observed to be 0.024 ± 0.005 [mg Nic/mg TPM] for product EC14-01.

Figure 6 shows nicotine yield per puff, ${\hat{Y}}_{Nic}$, as a function of flow rate and duration as defined by Equation (9). The shape of the surface is identical to ${\hat{Y}}_{TPM}$ because the nicotine mass ratio was observed to be independent of puff topography. For constituents not having a uniform mass ratio, *f_HPHC_*, the shape of the ${\hat{Y}}_{HPHC}$, surface shape will differ from that of TPM yield. The underlying data are illustrated by the markers, while the semitransparent surfaces reflect the 95% confidence interval, ε_YNic_. The value of Max ${\hat{Y}}_{Nic}$ was observed to be 0.634 ± 0.081 [mg/puff] for product EC14-01.

### 3.2. Summative Results for Quantitative Comparison of All 13 Products

The maximal response point of the emissions screening model for each of the thirteen products tested is shown in Table 3. The values of maximum TPM yield per puff varied widely, from 2.21 up to 90.94 [mg/puff], with similar variations in TPM mass concentration from 0.028 to 0.651 [mg/mL] and nicotine yield per puff from 0.101 to 4.175 [mg/puff]. The underlying data for model parameters are available in the Appendix A accompanying this article.

The emissions model screening results are presented graphically as interval plots in Figure 7. The results are sorted using the upper bound on the confidence interval for each parameter to compare each of the maximal emissions characteristics between products. This approach was chosen to illustrate the (i) combined effects of sample size and underlying product variation, (ii) desire to conduct comparisons using the most severe exposure scenario capable of being experienced by users of the product, and (iii) inform natural groupings for holistic “dashboard” comparisons between products. There are seven natural groups for Max *Y_TPM_*, six groups for Max *C_TPM_*, two groups for Max *f_Nic_*, and five groups for Max *Y_Nic_* for the family of thirteen products tested here. The results for product EC23-01 are off-scale for Max *Y_TPM_* and Max *Y_Nic_*.

Table 4 presents the lowest five a posteriori “between groups” statistically significant differences between maximal emissions characteristics presented in Figure 7 and denoted by the “+” markers. Only two significant levels for the aerosol mass ratio *f_Nic_* were observed among the products tested. As additional e-liquid concentrations are tested, the number of *f_Nic_* levels may be increased to reflect observed variation in this emission characteristic.

A proposed product emissions dashboard is presented in Figure 8. The dashboard is intended to provide an informational graphic which may be appealing to consumers, similar to dashboards about community pandemic risk levels, terror alert levels, or consumer product benchmark comparisons. Each row of the dashboard uses up to five color indicators reflecting the statistically significant levels of each factor presented in Table 4.

Observe that the Max *Y_Nic_* of products EC14-01, EC19-01, and EC22-01 was significantly larger than that of EC07-02, even though the aerosol nicotine mass ratio, *f_Nic_*, and branded nicotine concentration of the corresponding e-liquids were significantly lower. The relatively higher values of both Max *Y_TPM_* and Max *C_TPM_* associated with product EC14-01, EC19-01, and EC22-01 resulted in higher maximum nicotine yield to the mouth than product EC07-02. Users of those products will be exposed to significantly more TPM for a givn cumulative daily consumption of nicotine.

### 3.3. Association between ENDS Emissions and Product Characteristics

The next set of results investigated whether associations exist between maximum achievable emissions characteristics (what is delivered to the mouth of a user) and the underlying product characteristics (the design, composition, and operation of the ENDS). This information is essential for prioritizing proposed regulated product characteristics. As one exemplary analysis, multivariate linear regression was used to investigate possible associations between the maximum TPM yield per puff delivered to the mouth of a user, Max $\;{\hat{Y}}_{TPM}$, and the product design characteristics presented in Table 2. The results are illustrated in Figure 9.

The maximum yield of TPM per puff was significantly associated with ENDS power (*p* < 0.001) and was somewhat inversely related to the e-liquid nicotine concentration f_Nic_ (unpuffed), (*p* ≈ 0.056). Both associations are logical considering the physics of aerosolization. The presence of nicotine in the e-liquid solution tends toward increasing the saturation temperature of the mixture, while increased power to the coil enables more heat and mass transfer. Similar assessments investigating the dependence of Max ${\hat{C}}_{TPM}$, Max ${\hat{f}}_{Nic}$, and Max ${\hat{Y}}_{Nic}$ are presented in the Appendix A information accompanying this article.

The influence of each product characteristic on each adjusted emissions characteristic was assessed using added value plots, wherein the multivariate linear model was evaluated for each adjusted product characteristic while holding the remaining product characteristics constant. This resulted in a matrix of 24 responses; four emissions characteristics and six product characteristics. The Frisch–Waugh–Lovell theorem [15] results are shown in Figure 10.

Each panel of Figure 10 shows the linear regression best-fit association between one emissions characteristic (rows of the figure) and one product characteristic (columns of the figure). The discrete blue markers indicate the adjusted values of the emission and product characteristic while holding other product characteristics fixed. The solid line indicates whether an apparent positive or negative association exists, with the slope of the line indicated in each legend. The dashed red lines indicate the 95% confidence bounds on each association, and the black horizontal line indicates the mean of the adjusted emission response. If the black horizontal mean response line crosses the 95% confidence bounds, then the product characteristic may be taken to be a viable indicator or predictor of the emission characteristic. Conversely, if the mean response line falls between the 95% confidence bounds, then there exist an infinite number of associations which are statistically indistinguishable from the apparent association (the solid red line), and the product characteristic may not be taken to be a viable predictor of the emission characteristic. It is therefore asserted that the product characteristic of maximum nominal coil dissipation power, Max *P_Nom_*, is a positively associated predictor of Max *Y_TPM_* and Max *Y_Nic_* and a likely predictor of Max *C_TPM_*. The remaining product characteristics, of *f_Nic_*, *f_PG_*, *R_Coil_*, Min AD, and Max AD, taken individually, are not deemed to be viable predictors of emission characteristics.

## 4. Discussion

### 4.1. Key Findings of Regulatory Significance

Regulating e-liquid nicotine concentration alone is insufficient to limit the maximum yield of nicotine delivered to the mouth of a user. In fact, decreasing nicotine concentration in the e-liquid while keeping all other product characteristics fixed will result in a net increase in TPM exposure for a user who consumes a given mass of nicotine per day. That is, a consumer who compensates their behavior [16,17] to achieve desired nicotine consumption will increase their TPM exposure.

Limiting the maximum power permitted in vaping devices is an effective product characteristic to be considered for regulation. The maximum power capable of being dissipated in the coil of a vaping product has a statistically significant positive correlation with maximum TPM yield per puff, Max *Y_TPM_*, maximum TPM mass concentration in the aerosol, Max *C_TPM_*, and maximum nicotine yield per puff, Max *Y_Nic_*.

Rather than limiting the design characteristics of inhaled nicotine products, it may be far more effective to regulate the maximum permissible emissions from the product. As an analogy to environmental regulations, limits are typically placed on the amount and concentration of effluents (emissions) leaving a factory and going into the environment. It is generally left up to the manufacturer to determine how to achieve the emissions targets. Conversely, it is virtually impossible to anticipate the complex interaction between device and consumable characteristics and regulating those design characteristics provides numerous opportunities for product manipulation to “design around” the regulations. It is more effective to regulate the actual end-goal of emissions.

It is insufficient to evaluate emissions under a single operating condition as a basis for comparison between products or to establish substantial equivalence between products. Emissions from ENDS are dependent upon the puff flow rate, duration, e-liquid composition, device design, and operating power of ENDS.

It is recommended that US FDA tobacco product applications for marketing approval require experimental characterization of the emissions from the product over the entire operating range of the product: Min AD < d < Max AD and Min AF < q < Max AF at the maximum actual operating power of the product, Max P_Actual_.

Inhaled nicotine/tobacco product emission regulations should focus on direct emissions outcome measures in an effort to make regulations insensitive to product design manipulations. Traditional product characteristics considered for regulation include items such as e-liquid nicotine concentration and possibly coil resistance. However, such regulations may not achieve the desired public health outcomes. Even if ENDS manufacturers are constrained to a certain e-liquid nicotine mass concentration, they are able to manipulate numerous product characteristics to achieve a high nicotine yield per puff: increase the PCU de-energize duration (Max AD), decrease the coil resistance (*R_coil_*), increase the coil voltage or current, increase the coil power duty cycle (all manifest as Max P), decrease the ENDS flow path resistance, or modify the solvent saturation temperature (e.g., *f_PG_*). All of these manipulated product characteristics result in potentially adverse unintended public health consequences. We propose it is more effective to regulate the product characteristics of TPM (Max *Y_TPM_*) and nicotine (Max *Y_Nic_*) yield per puff. In the proposed case, manufacturers have free reign to manipulate numerous design parameters of their PCUs and e-liquids, but the end-result outcome measure remains consistently regulated.

### 4.2. Limitations and Scope

The results presented herein may be limited to pen- and pod-style ENDS not having user-adjustable power or flow paths. The screening emissions model, outcome measures, and method for assessing association between emission characteristics and product characteristics may be broadly applied to a variety of inhaled nicotine products including ENDS, combustibles, and heated tobacco products (also referred to as “heat not burn”). The methods may be extended to other electronic vaping products (EVPs) with further development.

This article has proposed a consumer-oriented dashboard (Figure 8) for quantitative comparison of ENDS devices and consumables. The dashboard may be combined with the list of product design characteristics (Table 2) as a recommended starting point for consumer packaging requirements. The approach to comparison of maximal product emissions (Figure 7) is recommended as a basis for quantifying substantial equivalence of tobacco/nicotine delivery products.

Similar to the proposed rule to limit the nicotine permitted in combustible cigarettes [18], reducing maximum allowable nicotine concentration in e-liquids may be suggested as a means of reducing the addictive potential of ENDS and the risk of initiating combustible cigarette use following use of ENDS [19] and reducing the probability of youth becoming newly addicted to nicotine [20]. The public health benefits of reducing nicotine dependence are well established [21].

However, for currently addicted users of nicotine, it is important to also assess the potential adverse unintended public health consequences which may be associated with increased TPM exposure arising from reduced nicotine concentration. The current work does not fully address the potential public health impacts of nicotine concentration regulation.

The relatively small number of unique power control units (PCUS) and reservoirs (pods or tanks) tested for each ENDS product, in conjunction with the limited number of repeated trials per flow condition, resulted in relatively large confidence intervals on the emissions surfaces. With additional trials, the confidence bounds may be reduced to permit a finer resolution comparison of maximal emissions between products. Even with the limited number of repeated trials reflected in the data sets presented, differences between products outweigh expected adjustments to the models which would result from collecting more data.

The maximum emissions characteristics (Max ${\hat{Y}}_{TPM},$ Max ${\hat{C}}_{TPM}$, Max ${\hat{f}}_{Nic}$, and Max ${\hat{Y}}_{Nic})$ reflect the severe exposure limits capable of being experienced by users of the product within the normal use operating envelope of the product. Two questions arise regarding the validity of relying upon the extrema for emissions comparison. The first question is: do the maximum values represent actual exposures which will be experienced by the user? A closely related question is: are there limits on the emissions model above which, while the device still operates, would never be observed by actual product users? The answers to both questions lie in characterizing the natural environment use patterns documenting how each product is actually used, as outlined in the next section.

### 4.3. Gaps Which Need to Be Addressed

A thorough understanding of user topography behavior in the natural environment is required to accurately model public health consequences of proposed product regulations. Studies to understand how users compensate both their short-term (puff and respiration topography) behavior and long-term (cumulative daily, weekly, and annual consumption) behavior in response to changes in product characteristics are needed. Studies are needed to investigate how users’ short- and long-term behavior changes: (a) as they become more or less addicted, (b) when dual-using nicotine/tobacco products, (c) in association with confounding factors such as alcohol use or illicit drug use, (d) with socioeconomic status, (e) among underrepresented groups, (f) among at-risk populations, (g) among those with mental health conditions, and (h) by pregnancy or lactation status.

A better understanding of the dynamic power control methods employed in ENDS will be valuable for establishing a first-principles causal relationship between ENDS operating power and emissions characteristics. Studies are needed to assess the effect of operating power and ENDS product characteristics which may impact maximum coil operating temperature, and hence give rise to chemical decomposition byproducts in aerosols.

Characterization of *f_Constituent_* for all compounds present in e-liquids and ENDS reservoirs and for all HPHCs identified in aerosols is needed. Studies are needed to fully identify and quantify all constituents in ENDS devices, consumables, and packaging. Studies of leach testing are needed to assess product degradation under normal and adverse storage and use conditions.

We recommend standardization of emission reporting outcome measures between laboratories as *Y_TPM_*, *C_TPM_*, and *f_Constituent_*.

## 5. Conclusions

Three emissions outcome measures (Y, C, and f) are recommended for adoption as standard quantities for emissions testing and reporting by manufacturers and research laboratories. Any two of the outcome measures permit calculation of the third, while reporting only one measure introduces ambiguity which may be exploited to circumvent emissions regulations.

A standard method for quantifying and comparing maximal emissions from ENDS products has been presented and demonstrated. Side-by-side comparison of the maximal emissions which can be delivered to the mouth of a product user provides a statistically robust basis for comparison of relative harm potential between ENDS. Requiring the emissions characteristics to be presented over the entire operating envelope of the ENDS ensures the full range of product use behavior is assessed when comparing ENDS for public health impacts.

A product design characteristics table has been proposed for comparative evaluation of ENDS devices and consumables. The product comparison table is intended for an audience of inhaled nicotine product regulators and researchers, serving as the basis for statistically robust assessment of substantial equivalence or lack thereof. Publishing an open access archive of such results for products already in the marketplace provides a basis for determining whether newly proposed products are likely to have a positive, neutral, or negative impact on harmful constituent exposure potential.

A product emissions dashboard has been proposed for comparative evaluation of ENDS exposure potential. The dashboard is intended for an audience of consumers without a statistical background, in a format familiar to individuals who have grown accustomed to seeing product comparisons for vehicle performance, nutrition labels, pandemic risk levels, and terrorist risk levels. The intent is to provide an adaptable, extensible dashboard which consumers can refer to when making product choices. Similarly, the dashboard may be helpful to clinicians who are counseling patients on strategies to reduce their nicotine dependence while being cognizant of harmful constituents they may be exposed to.

Maximum achievable power dissipated in the coil of ENDS is an effective regulatory parameter.

## Figures and Tables

**Figure 1 ijerph-19-02144-f001:**
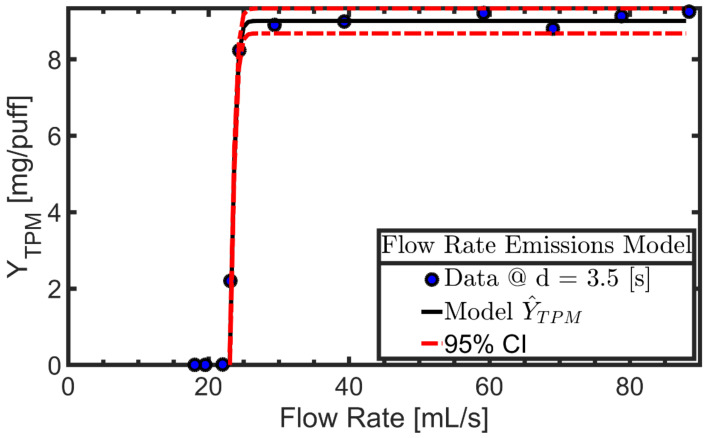
Product EC14-01. TPM yield per puff observed during “activation flow rate” screening trials conducted with nominal puff duration of 3.5 [s] and puff flow rate varying from 23 to 88 [mL/s]. The nonlinear regression emissions model (solid line) and 95% confidence interval (dashed line) are illustrated.

**Figure 2 ijerph-19-02144-f002:**
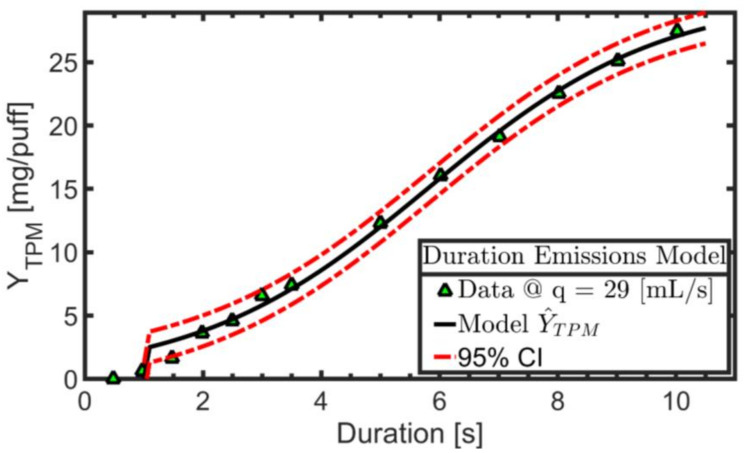
Product EC14-01. TPM yield per puff observed (markers) during “activation duration” screening trials conducted with nominal puff flow rate of 29 [mL/s] and puff duration varying from 1.0 to 10 [s]. The nonlinear regression emissions model (solid line) and 95% confidence interval (dashed line) are illustrated.

**Figure 3 ijerph-19-02144-f003:**
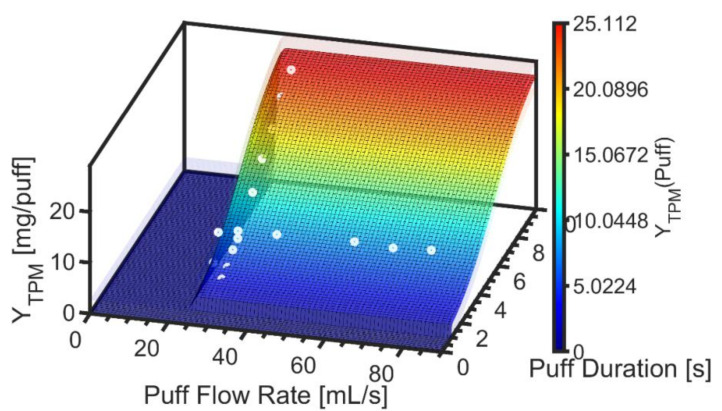
Product EC14-01. Screening emissions model for TPM yield per puff is illustrated as the colored surface plot, with semitransparent surfaces above and below the model surface to reflect the 95% confidence interval on the nonlinear regression. Underlying data are illustrated with markers.

**Figure 4 ijerph-19-02144-f004:**
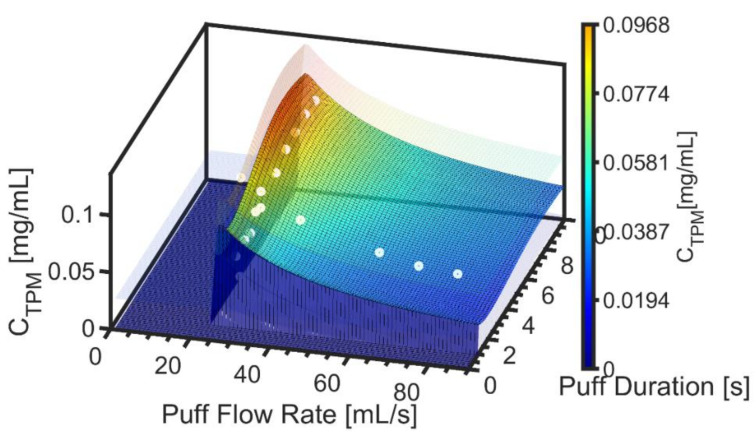
Product EC14-01. Screening emissions model for TPM mass concentration is illustrated as the colored surface plot, with semitransparent surfaces above and below the model surface to reflect the 95% confidence interval on the nonlinear regression. Underlying data are illustrated with markers.

**Figure 5 ijerph-19-02144-f005:**
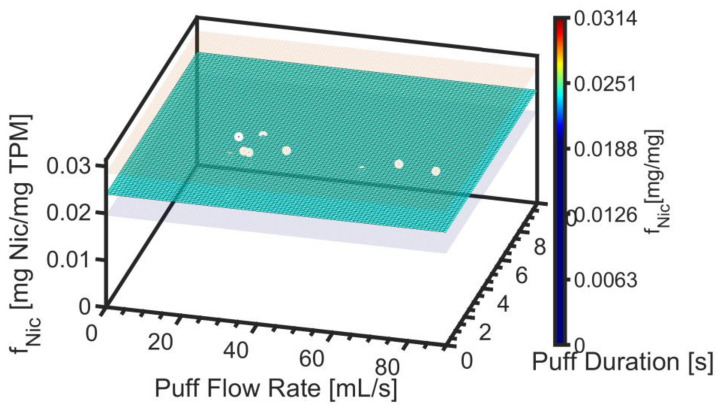
Product EC14-01. Screening emissions model for nicotine mass ratio is illustrated as the colored surface plot, with semitransparent surfaces above and below the model surface to reflect the 95% confidence interval on the nonlinear regression. Underlying data are illustrated with markers.

**Figure 6 ijerph-19-02144-f006:**
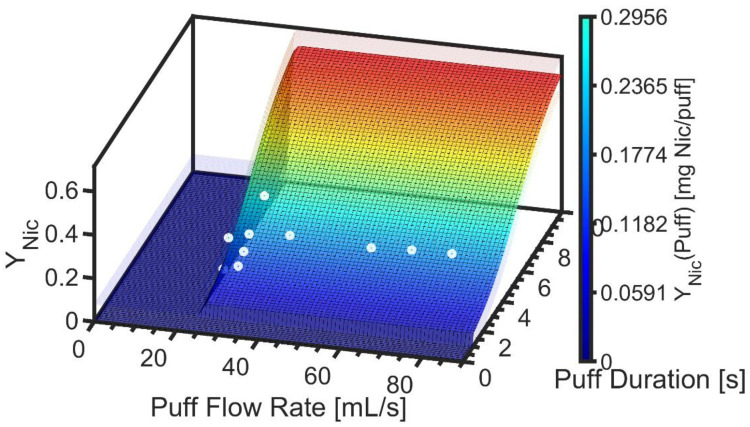
Product EC14-01. Screening emissions model for nicotine yield delivered to the mouth as a function of puff flow rate and duration is illustrated as the colored surface plot, with semitransparent surfaces above and below the model surface to reflect the 95% confidence interval on the nonlinear regression. Underlying data are illustrated with markers.

**Figure 7 ijerph-19-02144-f007:**
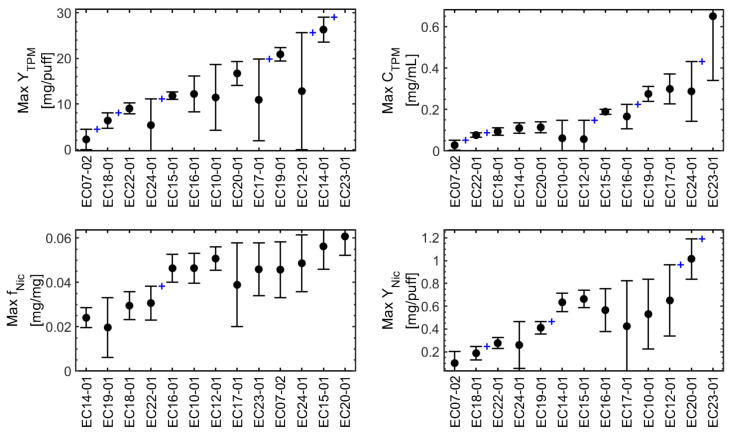
Interval plot for the maximal emissions characteristics demonstrating quantitative comparison between products. Results are sorted within each figure in order of increasing maximal value plus the upper 95% confidence bound from left to right. The “+” marker between successive data points indicates there is a significant difference between maximal product emissions (*p* < 0.05).

**Figure 8 ijerph-19-02144-f008:**
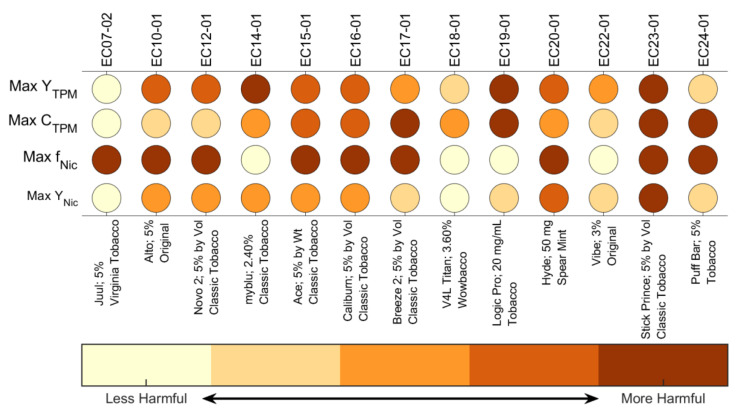
Maximal emissions screening dashboard for consumer-oriented comparisons between inhaled nicotine products. The horizontal axis indicates the ENDS device model, consumable labeled nicotine concentration, and flavor brand name for cross-reference to Table 2 and Table 3.

**Figure 9 ijerph-19-02144-f009:**
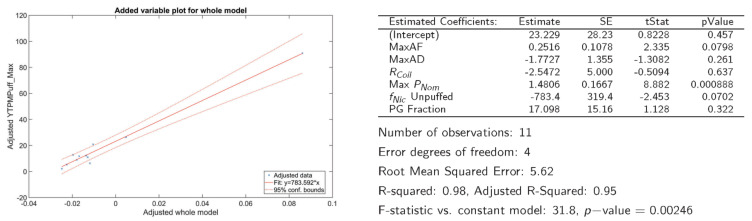
Assessing potential associations between Max ${\hat{Y}}_{TPM}$ and the product characteristics of *f_Nic_*_,_
*f_PG_*, *R_Coil_*, Max *P_Nom_*, MaxAF, and MaxAD.

**Figure 10 ijerph-19-02144-f010:**
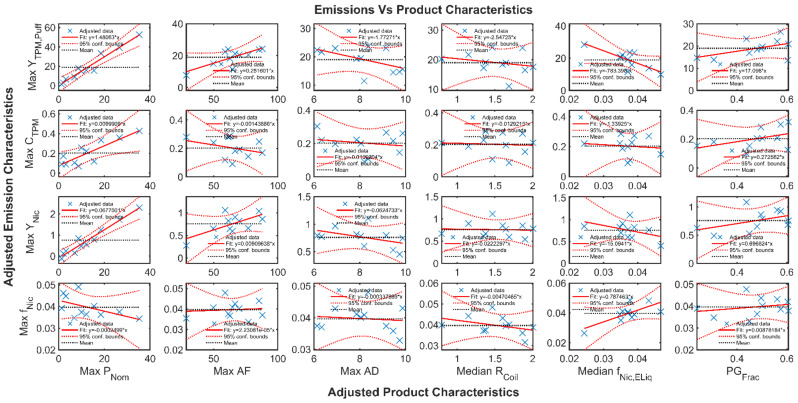
Added variable assessment of each adjusted emission characteristic (Max ${\hat{Y}}_{TPM}$, Max ${\hat{C}}_{TPM}$, Max ${\hat{f}}_{Nic}$, and Max ${\hat{Y}}_{Nic})$ as a function of each adjusted product characteristic (Max *P_Nom_*, MaxAF, MaxAD, *R_Coil_*, *f_Nic_*_,_
*f_PG_*) while holding other product characteristics constant. Thirteen pod- and pen-style ENDS devices are represented in the data set.

**Table 1 ijerph-19-02144-t001:** Son et al. [1] reported yield per puff with three repeated trials per condition for JUUL ENDS filled with Fruit Melody E-Liquid having a manufacturer-reported nominal nicotine concentration of 59 [mg/mL]. The underlying data reported by Son was extracted and converted into the nomenclature and units of the current article in order to provide a basis for comparison with results presented herein.

Nominal Puff Flow Rate [mL/s]	Nominal Puff Duration [s]	Nominal Puff Volume [mL]	TPM Yield [mg/puff]	St Dev [mg/puff]
25	2	50	2.00	0.20
25	3	75	2.10	0.20
25	4	100	2.80	0.20
25	5	125	3.30	0.30
33	4	132	2.80	0.20

**Table 2 ijerph-19-02144-t002:** Product design characteristics describing the 13 ENDS PCUS and corresponding consumable used for this investigation. Adapted from underlying data reported in [2].

Product ID	Device Manufacturer	Device Model	Consumable Manufacturer	Consumable Labeled Flavor	Unpuffed E-Liquid fNic [−]	PG Fraction [−]	Mean *Rcoil* [Ω]	Max *PNom* (*1*) [W]	MaxAF [mL/s]	MaxAD [s]
EC07-02	JUUL LABS	Juul	JUUL LABS	Virginia Tobacco	0.052	0.33	1.633	6.7	85	6.5
EC10-01	VUSE	Alto	VUSE	Original	0.052	N/R	1.063	10.2	50	5
EC12-01	SMOK	Novo 2	MAD HATTER JUICE	Classic Tobacco	0.039	0.44	1.463	7.4	58	8
EC14-01	BLU	myblu	BLU	Classic Tobacco	0.02	0.42	1.416	7.7	88	10
EC15-01	NJOY	Ace	NJOY	Classic Tobacco	0.05	0.48	1.034	10.5	58	5.5
EC16-01	UWELL	Caliburn	MAD HATTER JUICE	Classic Tobacco	0.039	0.44	1.405	7.8	88	10
EC17-01	ASPIRE	Breeze 2	MAD HATTER JUICE	Classic Tobacco	0.039	0.44	0.631	17.3	30	10.5
EC18-01	VAPOR4LIFE	V4L Titan	VAPOR4LIFE	Wowbacco	0.023	0.73	2.258	4.8	50	10
EC19-01	LOGIC VAPES	Logic Pro	LOGIC VAPES	Tobacco	0.016	0.77	2.443	4.5	48	12
EC20-01	LOONTECH	Hyde Original	LOONTECH	Spearmint	0.061	N/R	1.61	6.8	86	10
EC22-01	VUSE	Vibe	VUSE	Original Tobacco	0.03	0.23	2.693	4.0	68	6
EC23-01	SMOK	Stick Prince	MAD HATTER JUICE	Classic Tobacco	0.052	0.44	0.174	62.6	90	8
EC24-01	PUFF BAR	Puff Bar	PUFF BAR	Tobacco	0.049	0.54	1.688	6.5	50	3.6

Power was not experimentally measured in this investigation. Maximum nominal power was, for relative comparison only, computed from P_Nom_ = V^2^_Nom_/R_Coil_ where V_Nom_ was set to a constant value of 3.3 [VDC].

**Table 3 ijerph-19-02144-t003:** Maximal emissions characteristics for 13 pen- and pod-style electronic nicotine delivery systems.

Product	Device	Device	Consumable	Consumable	Max Y_TPM_		Max C_TPM_		Max f_Nic_		Max Y_Nic_	
ID	Manufacturer	Model	Manufacturer	Labeled	Value ±	95% CI	Value ±	95% CI	Value ±	95% CI	Value ±	95% CI
Code				Flavor	[mg TPM/puff]		[mg/mL]		[mg Nic/mg TPM]		[mg Nic/puff]	
EC07-02	Juul Labs	Juul	JUUL Labs	Virginia Tobacco	2.210	2.251	0.028	0.024	0.046	0.013	0.101	0.104
EC10-01	VUSE	Alto	VUSE	Original	11.433	7.230	0.061	0.085	0.046	0.007	0.531	0.305
EC12-01	SMOK	Novo 2	Mad Hatter Juice	Classic Tobacco	12.814	12.896	0.057	0.091	0.051	0.005	0.650	0.312
EC14-01	Blu	myblu	Blu	Classic Tobacco	26.341	2.707	0.110	0.026	0.024	0.005	0.634	0.081
EC15-01	NJOY	Ace	NJOY	Classic Tobacco	11.790	0.829	0.190	0.012	0.056	0.010	0.663	0.076
EC16-01	Uwell	Caliburn	Mad Hatter Juice	Classic Tobacco	12.193	3.960	0.167	0.059	0.046	0.006	0.565	0.186
EC17-01	Aspire	Breeze 2	Mad Hatter Juice	Classic Tobacco	10.911	8.972	0.300	0.072	0.039	0.019	0.424	0.401
EC18-01	Vapor4Life	V4L Titan	Vapor4Life	Wowbacco	6.365	1.664	0.094	0.019	0.030	0.006	0.188	0.061
EC19-01	Logic Vapes	Logic Pro	Logic	Tobacco	20.911	1.445	0.276	0.036	0.020	0.013	0.412	0.054
EC20-01	Loontech	Hyde	Hyde	Spear Mint	16.714	2.646	0.114	0.027	0.061	0.009	1.015	0.177
EC22-01	VUSE	Vibe	VUSE	Original	8.998	1.211	0.077	0.010	0.031	0.008	0.276	0.048
EC23-01	SMOK	Stick Prince	Mad Hatter Juice	Classic Tobacco	90.943	25.939	0.651	0.312	0.046	0.018	4.175	1.688
EC24-01	Puff Bar	Puff Bar	Puff Bar	Tobacco	5.343	5.721	0.288	0.144	0.049	0.013	0.260	0.204

**Table 4 ijerph-19-02144-t004:** The lowest five statistically significant differences between maximal emissions characteristics observed across the thirteen product combinations tested.

Factor	Max *Y_TPM_* [mg/puff]		Max *C_TPM_* [mg/mL]		Max *f_Nic_* [mg Nic/mg Nic]		Max *Y_Nic_* [mg/puff]	
Level	Lower	Upper	Lower	Upper	Lower	Upper	Lower	Upper
1	0.0	4.46	0.0	0.052	0.0	0.038	0.0	0.249
2	4.46	8.03	0.052	0.087			0.249	0.465
3	8.03	11.06	0.087	0.148			0.465	0.962
4	11.06	19.9	0.148	0.226			0.962	1.192
5	19.9	inf	0.226	inf	0.038	inf	1.192	inf

## Data Availability

Data supporting the results reported herein is provided in the Appendix A associated with this article. Additional underlying predicate data reported as Appendix A accompanying reference [2].

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
