# Peer review of "Proposed Standard Test Protocols and Outcome Measures for Quantitative Comparison of Emissions from Electronic Nicotine Delivery Systems"

_ijerph, 2022, doi:10.3390/ijerph19042144_

Round 1

Reviewer 1 Report

On the 1 January 2022

Review of article: Proposed Standard Test Protocols and Outcome Measures for 2 Quantitative Comparison of Emissions from Electronic Nico-3 tine Delivery Systems. by Edward C. Hensel et all

General remarks

  • This paper is confusing, without clear purpose.

You must read reference 2 (by the same authors) to understand the approach of the authors in this article.

Even after reading Reference 2, one does not fully understand the purpose of the modelling proposed.

  • If this is to say that when one does not inhale on the e-cigarette there is no aerosol coming out, it seems a little complicated calculation for obviousness without practical use. A user quickly realizes (as he did on his cigarette that if you inhale hardly any, there is no smoke coming in and that if you inhale harder there is more smoke coming in.

  • For maximum emissions, the aerosol emission at the tip of the electronic cigarette reaches a maximum when the air flow through the electronic cigarette allows to deliver all of what is produced at a present moment. Once again, the figure which could have been technically interesting to characterize the product would have been the amount of production of vapor by the resistance, rather than the result of the washing by the air of the chamber.

Two of the three aim of the study are announced to be

  • “Quantifying and comparing maximal emissions from ENDS products”
  • “Establishing comparative equivalence of maximal emissions from ENDS”

The paper doesn’t present the interest of measuring the practical emission. This king of measurement is very unusual for tobacco product and form the same regular cigarette a smoker may receive 0,6 to 6 mg of nicotine according to the way of use of e-cigarette. All person who are nicotine dependant know how to use tobacco and tobacco related product. If the delivery on one device is two small, the user makes the decision to take more puff. Reading the paper, the reviewer cannot understand the interest of such measure who may have only an interest who want to put a new product on the market and who don’t need a precise assessment. à So, the interest of the paper is in any case very limited and the modelisation add nothing to the previous one.

Note that users who want high powers adjust (if available) the electrical power applied to a well-defined resistance surface and on the air admission into the chamber and not the simple washing by the air from the nebulization chamber.

  • The direct inhalation of e-liquid reported by the authors exists in vaping machines when the position of the cigarette is inadequate, placed horizontally or above the “mouth” (I did not find any precision on the position of the e-cigarettes in the first study or in the studies cited which does not allow the measurements described to be reproduced, which is non adequate for a scientific publication). Note than the French AFNOR standards published in 2015, in addition to specifying the conditions and topography of puffs in the vape machine, recommend an inclination of 45 °, which eliminates this direct inhalation.

NB: If a user tries to use an electronic cigarette, for example when a cigarette user is lying on his bed, this phenomenon exists, but the bad taste of the first drop of e-liquid leads him to immediately stop this practice (without health risk for some drops although very unpleasant). In practice there is no significant risk, but indeed it could have been specified in the first article (ref 2) in the recommendations to users never to use the e-cigarette in a declining position of the mouthpiece in relation to the rest of the e-cigarette (e.g., when you are lying down).

  • For emissions of interest from e-cigarettes, the article is confused and don’t answer to the question and why the emissions must be assessed. The authors should have clearly mentioned 3 different problems:

1- The good delivery of nicotine which is the substance need for smokers to quit cigarettes and nicotine addiction. A perfect compensation of the nicotine needs to fill the innumerable nicotinic receptors present in addicted smokers by a gentle delivery repeated throughout the day as do patches and other oral nicotine replacement therapy. Nicotine which is the cause of nicotine addiction while 90% of users who switch from tobacco cigarettes to vaping see their daily nicotine needs drop after the last cigarette by about 1% per day, 30% per month, thus helping to get out of the nicotine addiction induced by tobacco.

2- The absence of release of undesirable substances under normal conditions of use. It must be clearly specified what we are trying to control to preserve the health of the user.

AFNOR XP90-300-3 standards, for example in France, require that 300 puffs delivered in the laboratory under standard conditions (puffs of 2 seconds and 55 ml, etc. deliver an amount of formaldehyde and other aldehydes lower than what is required for 24 hours on indoor air quality, a more protective for diacetyl than the workplace standards reported at 24 hours (moreover diacetyl cannot be an ingredient according to these standards), a metal emission lower for metals than what is authorized for inhaled drugs (always over 24 hours)

  • Note that the number of PM which is the centrum of this article is meaningless because it is very mainly droplets and not solid particles. (The toxicity of solid PM2.5 is high because of physical effect, toxicity of liquid droplet 2.5mcg is much lower if the liquid contains only low irritants and toxic compounds. Remember that the principle of vaporizing an e-liquid only produces gases when the e-liquid is in contact with the heated electrode. The gases will re-condense into droplets which will be inhaled (with a half-life before returning to the gaseous state of 11 seconds for PG liquids and 28 seconds for Glycerine Liquids). The release of solid particles from the electrodes or wicks is minimal with modern systems and represents only a tiny fraction of the emissions of solid particles, which are present at a concentration of solid particles close to those in ambient air (except error in design or manufacture of devices or use of overheating devices in the absence of e-liquid or by applying disproportionate electrical power to the resistance).

Note that neither the article to be revised nor reference 2 of the submitted paper clearly considers the surface of the coils.

The power is designated as an important point (wattage or voltage imposed on the resistance, but this power does not mean anything if we do not know the resistance (ohm) and the surface of the coil in contact with the e-liquid (mm2) It is recalled that, for example, the same fire from a gas stove will be considered too powerful if a tiny saucepan is placed (which will burn and then lead to the emission of toxic substances) or very insufficient to vaporize if it’s' A very large pot, not allowing the liquid placed in the pot to come to a boil, while the fire (power) and the material of the pot were the same in both cases. talking about resistance in Ohm and the surface of the coil exposed to liquids does not make it possible to define an emission (Unless the goal is to compare only e-cigarettes of the same batch and same brand whose resistance (ohm and surface in contact with e-liquid is identical), but does not allow e-cigarettes to be compared with each other

Furthermore

  • On devices analysed:

The measures only seem to concern single-use (pen) or cartridge electronic cigarettes but not refillable ones, although in some parts of the article it seems that MODs are considered, but it is not clear how.

  • The recommended Cambridge filters are well suited for the filter collection of cigarette smoke consisting almost exclusively of solid particles. Vape emissions consist almost exclusively of liquid droplets and when you pass an aerosol produced by 50 puffs of up to 71ml the filter is fully saturated. Authors say having verified that the weight loss of the electronic cigarette (or the tank) was identical to the weight collected on the filter, but I could not identify in the article the numerical values ​​for the 13 e-cigarettes, to demonstrate that there was no loss of mass and therefore confirm your assertion. Positioning a second trapping system downstream of the filter is another method of verifying that the Cambridge filter has collected all particles.

  • Authors speak, as the United States Tobacco Administration does to refer to the ENDS, but in Europe and in many countries’ tobacco products are products that contain tobacco (this makes some sense in a rather scientific article than to designate as tobacco products that are not tobacco!). ENDS contain only pharmaceutical grade nicotine most often which is no more a tobacco product than nicotine patches or gum. It should say at least "classified as a tobacco product by the FDA" or "product assimilated to tobacco products".

  • The interest of modelisation if very poor, for producer and for users (as the main concern is the adequation of power to the coil characteristic (ohm and surface in contact with the liquid).

There is no information on the use of ENDS with a lack of e-liquid in the tank who produce overheating and emission of aldehydes and others toxic substances.

  • The air flow in the tank is not the major concern for users.

The only useful information is that the nicotine is delivered in the same proportion than in the liquid: if the liquid contains 2% nicotine, the aerosol contains 2% nicotine and the weight lost of e-cigarette is due to 2% nicotine.

In details

I renounce to reproduce my detailed analysis because the fundamental observation in not compatible with the publication of the paper as it was presented.

Reviewer 2 Report

I have read the content of the article entitled "Proposed Standard Test Protocols and Outcome Measures for Quantitative Comparison of Emissions from Electronic Nicotine Delivery Systems". This is an interesting study with very broadly described research results and numerous figures and tables. However, it is worth for the authors to expand the conclusions section, which is disproportionately short in relation to the rest of the article, in particular the results section.
In the description of the results, it is also worth refining the designation and placement in the text of Figures 3 to 6, which have no references in the text.

Reviewer 3 Report

This study compared 13 ENDS products in terms of emission characteristics and provided useful information for ENDS regulation. In general, this manuscript is completed, but the language can be more succinct and use active tone to make it more readable. I have the following suggestions:

1. Two “table 2” and several “figure 2”

2. Did not explain the abbreviations used in tables and figure, make it hard to read: such as “MAX P Nom”, and did not spell full name for TPM.

3. Lines 80-85 mentioned three main outcome measures: YTPM, CTPM, fNic, however, later there were 4 outcome measures listed in figure 7 (between line 34-335).

4. In figure 8, the level of harmfulness seems objective, did it come from a formula?

5. In table 3 (below line 323), it is better to included the p value, R2 in the table, instead of list this information in the supplemental tables.

6. The background can be organized better with providing brief review of current ENDS regulation, then what outcome measures used in Son et al.’s study. Further, author should explain why they think these three outcomes are more important: YTPM, CTPM, fNic, 

7. In the background 1.2 section, author should state how the three main outcomes related to the product harmfulness. 
